# HEDGE YOUR ACTIONS: FLEXIBLE REINFORCEMENT LEARNING FOR COMPLEX ACTION SPACES

## ABSTRACT

Real-world decision-making is often associated with large and complex action representations, which can even be unsuited for the task. For instance, the items in recommender systems have generic representations that apply to each user differently, and the actuators of a household robot can be high-dimensional and noisy. Prior works in discrete and continuous action space reinforcement learning (RL) define a retrieval-selection framework to deal with problems of scale. The retrieval agent outputs in the space of action representations to retrieve a few samples for a selection critic to evaluate. But, learning such retrieval actors becomes increasingly inefficient as the complexity in the action space rises. Thus, we propose to treat the retrieval task as one of listwise RL to propose a list of action samples that enable selection phase to maximize the environment reward. By hedging its action proposals, we show that our agent is more flexible and sample efficient than conventional approaches while learning under a complex action space. Website.

## 1 INTRODUCTION

An essential goal of reinforcement learning (RL) is to solve tasks humans cannot solve, such as those with innumerable action spaces. For instance, recommender systems have a set of millions of discrete items, and robotic control requires acting in continuous dimensions. In such domains, all the actions cannot be enumerated feasibly to perform RL. Thus, actions are associated with continuous parameterizations or representations that enable generalized learning. We consider continuous and large parameterized discrete action spaces as problems of innumerable actions.

Typically innumerable action space tasks are solved in two phases: retrieval and selection (See Fig. 1). QT-Opt (Kalashnikov et al., 2018) learns robotic manipulation by retrieving actions from a distribution fitted using the cross entropy method and selecting the action that maximizes a learned Q-function. Similarly, Dulac-Arnold et al. (2015) performs large discrete action RL by acting in the space of action representations, retrieving k-nearest-neighbors, and selecting with a Q-function. Continuous actor-critic approaches like DDPG and SAC (Lillicrap et al., 2015; Haarnoja et al., 2018) directly learn an actor with a learning objective of retrieving an action that maximizes a Q-function.

However, action retrieval in these approaches takes the form of a single action, a distribution, or a local neighborhood defined on the action space. While these forms of retrieval work when the selection Q-function is smooth over the actions, it becomes a limiting factor in complex action spaces. The action space can be imprecise, noisy, high-dimensional, or independently derived from the task (such as in recommender systems), leading to a mismatch between the action representations and their effects on the task. Therefore, to enable efficient reinforcement learning in complex and innumerable action spaces, we aim to address the goal of performing robust action retrieval.

Our critical insight is to perceive the problem of action retrieval as one of listwise reinforcement learning (Sunehag et al., 2015). Concretely, the retrieval network is considered an RL agent with a modified action space of picking a list of $k$ action candidates, trained to enable the selection Q-function to maximize the environment reward (Fig. 1). Listwise retrieval improves the efficiency of RL in two ways. Firstly, retrieval lags behind selection during initial training because it is trained with RL loss. By hedging or diversifying the retrieved actions, our selection phase gets better candidates to maximize, enabling directed exploration of the task. Second, listwise retrieval can learn to adapt the list composition (e.g., diverse or similar) over training because the actions are not

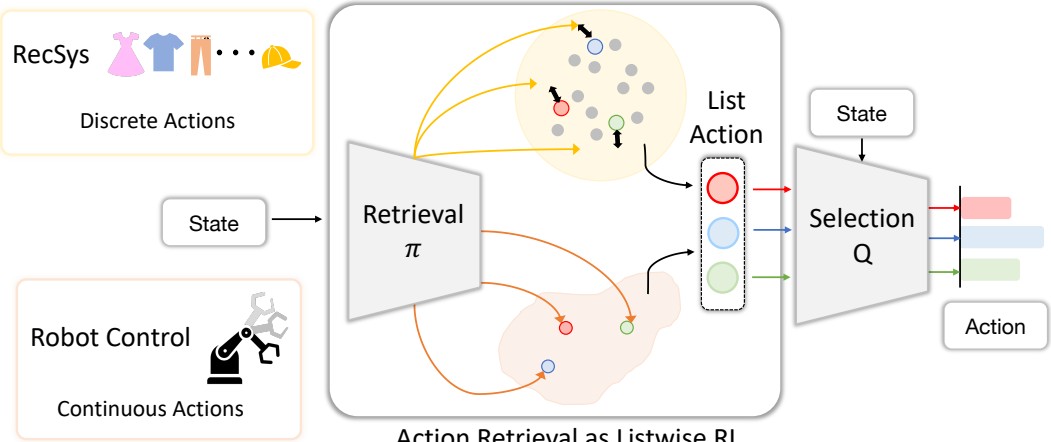

Figure 1: Large discrete action set and continuous action space tasks cannot be solved with a single Q-function because it is infeasible to enumerate all the actions. A common approach is to (a) retrieve $k$ actions, and (b) select only from those actions. We posit that the retrieval task can be generally seen as listwise RL, where the retrieval agent must learn to output the list of candidates as a whole. This enables flexible learning in complex innumerable action space RL.

constrained to a predefined distribution or local neighborhood form. We show that the flexibility of listwise retrieval is crucial in complex action spaces.

To this end, we propose a novel framework FLAIR, Flexible Listwise ActIon Retrieval, that can incrementally build a list of $k$ candidate actions without enumerating all possible list combinations. We extend cascaded DQN (Chen et al., 2019c; Jain et al., 2021) to continuous action space resulting in a cascaded DDPG framework with $k$ actors and $k$ critics. Specifically, an actor outputs a candidate action for each list index while considering the state and the partially built list as input. Each cascaded actor is trained to maximize its associated critic's value, while the critics are trained to maximize their environment reward. Overall, each actor-critic pair learns to retrieve a candidate action that can optimize the environment reward when combined with the current list of candidates.

Our primary contribution is introducing the problem of complex innumerable action spaces in reinforcement learning. We make the retrieval-selection approach flexible by incorporating listwise RL. We demonstrate our proposed method FLAIR learns to perform listwise action retrieval that enables flexible and efficient decision-making in innumerable action space tasks, such as recommender systems, a novel mine-world environment, and continuous control.

## 2 RELATED WORK

### 2.1 LARGE DISCRETE ACTION SPACE RL

Continuous action space is a common problem in RL, with several on-policy algorithms like TRPO (Schulman et al., 2015), PPO (Schulman et al., 2017), and off-policy algorithms like DDPG (Lillicrap et al., 2015), SAC (Haarnoja et al., 2018) proposed for it. Dulac-Arnold et al. (2015); Chandak et al. (2019) address the issue of large discrete action spaces in reinforcement learning, by learning an actor-critic framework in the space of action representations. Likewise, several approaches are based on the retrieval and selection framework (Tan et al., 2019; Chen et al., 2019b; Tessler et al., 2019; Narasimhan et al., 2015). We show that such conventional approaches suffer in complex action spaces due to their predefined constraints on the form of action retrieval.

Recently, Hubert et al. (2021) propose model-based planning in innumerable action spaces. Their insight of using a retrieval and selection framework is similar to ours, to avoid enumerating all possible actions. They propose an important sampling-like loss modification to enable learning with a subset of samples. However, unlike our work, this work focuses on the selection network, while their retrieval network is assumed to be specified as a proposal distribution and not learned. (Van de Wiele et al., 2020) learn a proposal distribution that can do $\arg\max Q$ by supervised learning on targets obtained from a stochastic maximization procedure on the Q-function. However, this stochastic

search procedure is susceptible to the action space's complexities. Chen et al. (2019a) train a fixed tree-structured policy network that reduces Q-function evaluation to logarithmic complexity in the number of actions. However, this cannot support generalization in the action representation space is thus inefficient and unscalable.

## 2.2 LISTWISE ACTION SPACE

Listwise RL has a combinatorial action space in the list size. Also known as slate RL, it has applications in recommender systems (Sunehag et al., 2015; Ie et al., 2019b; Chen et al., 2019c). In recent work, Jain et al. (Jain et al., 2021) proposed a generalizable policy architecture to perform better listwise RL in varying action spaces by considering interdependencies of the available actions as well as items within a list. Their architecture stands for Cascaded DQN(CDQN; Chen et al. (2019c)), where a set of Q-networks sequentially build a list while passing the intermediate list information onto the next Q-network to work out the action at the following list index. However, CDQN is based on a discrete action space DQN, and thus, unscalable to innumerable action spaces. In this work, we propose how to perform cascaded listwise RL in continuous action space to achieve action retrieval for innumerable action space tasks.

## 2.3 COMPLEX ACTION SPACE

A common form of complex action space arises in tasks that learn action representations independently from the task. Tennenholtz & Mannor (2019) use expert demonstrations to extract action representations based on co-occurrence of actions. Jain et al. (2020) propose an unsupervised framework to obtain action representations by encoding sets of action observations. While representation learning is useful for reducing the complexity of action space, it is conventionally done in a task-independent manner, which results in potentially challenging representations for action retrieval.

## 3 PROBLEM FORMULATION

A crucial goal of RL is to solve decision-making tasks that humans cannot solve, such as recommending the appropriate item to a user from a catalog of millions of items. Such problems become incredibly challenging when an agent's and environment's action interface is complex. For instance, complex robots require high-dimensional control and real-wold applications can have environments where a large stochastic noise is applied to an agent's actions. In RecSys, the action representations often come from task-independent domains, such as unsupervised learning on general item characteristics (Jain et al., 2021; 2020). Generally, we define complex action spaces where the representations of actions available to the agent are not directly amenable to the task it is solving.

### 3.1 RL WITH INNUMERABLE ACTION SPACE

We consider a unified setup for RL with large discrete action sets and continuous action spaces. Specifically, our task follows a Markov Decision Process (MDP), defined by a tuple $\{\mathcal{S}, \mathbb{A}, \mathcal{T}, \mathcal{R}, \gamma\}$ of states, actions, transition probability, reward function, and a discount factor, respectively. When the action space $\mathbb{A}$ is continuous, it has a $D$-dimensional continuous parameterization, $c_a \in \mathbb{R}^D$. For discrete action sets such that $|\mathbb{A}| >> 1$, we use $D$-dimensional action representations $c_a \in \mathbb{R}^D$ to denote an action $a \in \mathbb{A}$. This encodes the characteristics of the action accessible to the agent, for example, a compressed representation of image and text features for each item in recommender systems. Furthermore, at each time step $t$ in the episode, the agent receives a state observation $s_t \in \mathcal{S}$ from the environment and acts with $a_t \in \mathbb{A}$. Then, it receives the new state after transition $s_{t+1}$ and a reward $r_t$. The objective of the agent is to learn a policy $\pi(a|s)$ that maximizes the expected discounted reward over evaluation episodes, $\mathbb{E}_\pi \left[ \sum_t \gamma^{t-1} r_t \right]$.

### 3.2 RETRIEVAL AND SELECTION FRAMEWORK

Value-based methods such as Q-learning (Watkins & Dayan, 1992; Mnih et al., 2015) enable efficient reinforcement learning by being suitable for off-policy training. However, for innumerable action spaces, acting and training Q-functions becomes infeasible due to its reliance on computing $\arg\max_{a \in \mathbb{A}} Q(s, a)$. To make it feasible, prior works learn a two-stage framework 1. A retrieval

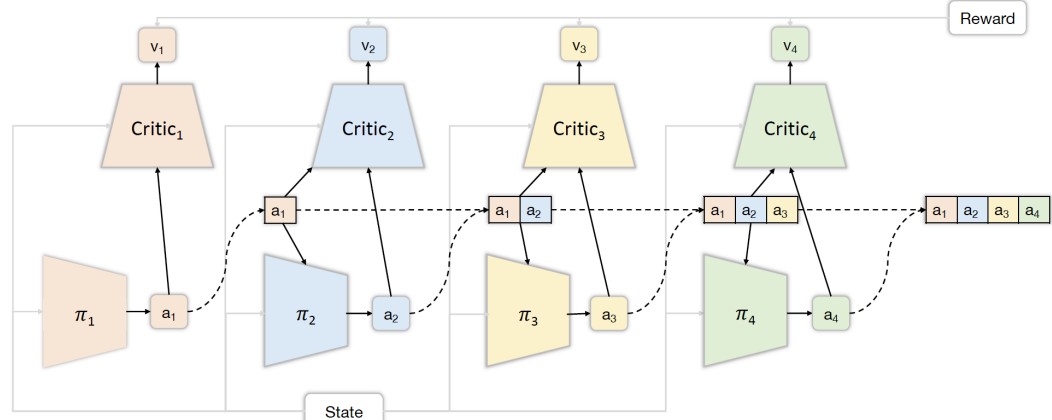

Figure 2: FLAIR employs a cascaded DDPG architecture to generate listwise actions as the retrieval agent. Each actor $\pi_i$ outputs a continuous action $a_i$ while taking in the state and currently built list $[a_1, ...a_{i-1}]$ as input. Likewise, each critic $Q_i$ learns to compute the episodic return based on the state, current list, and the policy's action. The cascaded DDPG architecture of retrieval is trained in conjunction with the selection Q-function.

stage obtains a small subset of candidate action(s), which the Q-function is evaluated over in the selection stage. For discrete action space, there is an additional step of finding $k$-nearest neighbors in the action representation space to retrieve actual discrete actions, based on a similarity measure such as Euclidean distance (Dulac-Arnold et al., 2015). In prior work, QT-Opt (Kalashnikov et al., 2018) retrieved Q-function maximizing actions using a distribution fitted via cross entropy method, while actor-critic approaches train a deterministic actor in DDPG (Lillicrap et al., 2015) or a stochastic actor in SAC (Haarnoja et al., 2018) to maximize the Q-function directly.

### 3.3 CHALLENGES OF ACTION RETRIEVAL IN COMPLEX ACTION SPACES

Action retrieval is challenging in complex action spaces when the manifold of action representations is difficult to act and learn. In simple RL tasks, the optimal policy $\pi(a|s)$ is relatively smooth over the action space. Thus, $\hat{a} = \pi^*(a|s)$ and $\hat{a} + \delta a$ are expected to have similar behavior in the task for small values of $\delta a$. Thus, prior retrieval agents relying on $k$-nearest-neighbors over discrete action representation queries have a wide margin of error regarding action selection. Similarly, continuous actor-critic approaches that output a point action query or a distribution (usually Gaussian) over the action space can learn well.

However, in RecSys or high-dimensional noisy robotic control, nearby actions $a$ and $a+\delta a$ may have different environmental effects. This is because the action representation space may not correspond to the task, and the relevance of actions can even vary depending on the state. For example, a laptop and a book may be similar or very dissimilar for different users. Therefore, in such complex innumerable action spaces, prior methods which assume a predefined structure of outputting over the action space (like k-NN, smoothness, similarity local neighborhood) are more challenging and inefficient to learn.

## 4 APPROACH

Our goal is to design an action retrieval framework robust to complex action spaces to address the challenges in Sec. 3.3. Our central insight is that the problem of retrieval of action samples is equivalent to reinforcement learning in a listwise action space (Sec. 4.1. To enable listwise RL in the continuous space of action representations, we adapt prior work on listwise DQN (Chen et al., 2019c; Jain et al., 2021) by learning to incrementally build a list of actions using cascaded actor-critics (Sec. 4.2).

### 4.1 FLAIR: FLEXIBLE LISTWISE ACTION RETRIEVAL

We propose FLAIR, a listwise RL framework for learning to retrieve action samples for RL with complex action space. Listwise RL supports directed exploration during initial training by enabling the selection Q-function to maximize the environment return better. Further, it allows the actor to be more robust to complex action spaces by having more flexibility in the actions it can sample. Together, by learning to sample a list of actions, our method can enable more sample efficient RL in discrete and continuous complex action spaces.

### 4.2 CASCADED ACTOR-CRITICS FOR CONTINUOUS LISTWISE RL

We require a continuous listwise RL agent to sample a list of action representations from the action space. Listwise RL itself is a combinatorially complex problem because there can be a large number of lists that can result from various action combinations. In conjunction with the innumerable action space, this makes listwise RL challenging. To tackle this problem in discrete action spaces, (Chen et al., 2019c) proposed Cascaded DQN where $k$ cascaded Q-networks learn to incrementally build a list of $k$ candidate actions.

In this work, we extend the idea of cascaded networks to continuous action space listwise RL (Fig. 2). We propose to use cascaded actor-critics that learn to output a constant action for each list index. Each actor-critic pair is tasked to learn to retrieve an action sample complementary to the currently built-up list input to maximize the environment reward.

**Policy Architecture**: We choose Deep Deterministic Policy Gradients (DDPG) (Lillicrap et al., 2015) as our base continuous action space algorithm, but any continuous action space approach can be extended to cascaded architectures. Specifically, the $i^{\text{th}}$ critic Q-function is trained to maximize the environment reward while being conditioned on the list built upto $(i-1)^{\text{th}}$ index. Subsequently, the $i$ actor is trained to output an action that maximizes the $i^{\text{th}}$ critic, while conditioned on the current list. The actor is trained by backpropagating gradients through the action to the critic, as in DDPG. All actors and critics share a list encoder that is flexible to the size of actions. For our experiments, we use Deep Set (Zaheer et al., 2017) as the choice of list encoder. More details on the policy architecture and implementation are present in Appendix B.

## 5 ENVIRONMENTS

We evaluate ARDDPG on the following three complex action space scenarios where the agent learns to act in the following task that each of which is equipped with the complex action space; (1) Tools reflecting the breakable object in a 2D Grid World to shorten the optimal path to the goal, (2) Items reflecting the characteristic of items to recommend items relevant to user, and (3) Actions reflecting the rotation of joints of a robot to perform task. Figure 3 provides an overview of the tasks, the action space construction, and an illustration of the expected behavior. More environment details such as tasks, action representations, and data collection are present in Appendix A.

### 5.1 MINING TASK: MINE WORLD

We modify the grid world environment (Chevalier-Boisvert et al., 2018) where an agent navigates a 2D maze while removing mines by appropriate pick-axes to reach a goal as soon as possible. As the action space, the agent always has access to 4 direction movements, which are going up, down, right and left. Further, the agent can use one of the many pick-axe actions to remove mines that are placed in the maze randomly. The digging of mines can result in breaking the mine or another mine, depending on the tool and the type of mine. Given the full observation of the maze, the agent needs to learn to navigate to the goal as soon as possible. However, sometimes there will be some mines blocking the way, so the agent also need to apply correct tools to break the mines. Successful digging and reaching the goal yield rewards to the agent. Each action is associated with a domain-defined action representation. We make the complex action space, by introducing Gaussian noise in each dimension. So, the manifold of the action representation space is noisy and imprecise.

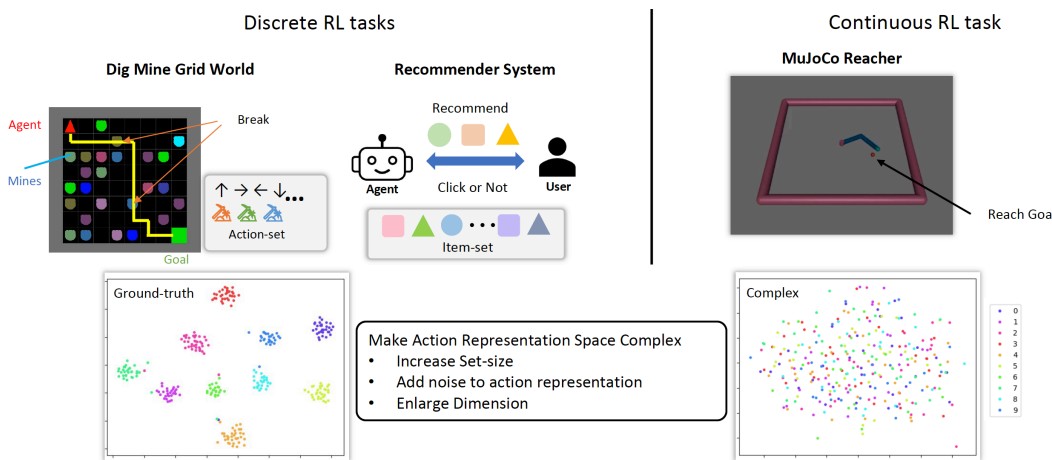

Figure 3: **Environment Setup.** (Left) In Mine-World, the red agent must remove mines to reach the green goal. (Middle) In simulated recommender system(RecSim), the agent selects a item given the set of items. (Right) In Continuous control tasks, the agent sends the right control signals to the robot to reach the goal.

## 5.2 SIMULATED RECOMMENDER SYSTEM: RECSIM

Recommender system (RecSys) is a natural application of large discrete action space RL - for instance, there can be a large number of news articles or videos are recommendable every day. The agent needs to recommend a right item out of a large set of items given the user preference information from the environment. We use RecSim (Ie et al., 2019a) to simulate user interactions and equip it with the larger set of items. In the item-set there are two sizes of the item-set 10000 and 5000 items. We assume a fully observable environment with the state as the user preference vector and the action representations as item characteristic. So, the agent needs to find the most relevant item from the large item-set based on the given the user preference information at each time-step.

## 5.3 CONTINUOUS CONTROL: REACHER

The 2D Reacher MuJoCo environment (Todorov et al., 2012) is a simple two-link planar robot tasked to reach a randomized target location, originally with a 2-dimensional action space. We modify the action space to become complex and 50-dimensional. Specifically, the 50-D action space accessible to the agent, goes through a non-linear matrix transformation to result in the 2-D action space. Further, we introduce a binary state variable that randomly switches one of two matrix transformations at the start of every episode. While the transformation is still smooth, this environment presents a demonstration of how a simple environment can also become complex when the action space is unsuited for the task.

## 6 EXPERIMENTS

We design experiments to answer the following questions in the context of RL with complex innumerable action spaces: (1) How effective is listwise retrieval in Complex Large action space compared to conventional forms of retrieval? (2) How do various action retrieval methods scale with different complexities of action representations? (3) Is our method sensitive to the choice of list encoder? (4) Does incorporating listwise metrics like diversity improve the performance of FLAIR?

### 6.1 EFFECTIVENESS OF LISTWISE RETRIEVAL IN COMPLEX LARGE ACTION SPACE

We evaluate our proposition against the following baselines, ablations, and variations of our method to validate the effectiveness of listwise retrieval.

### 6.1.1 BASELINES AND ABLATIONS

- **DDPG k-NN**: follows Dulac-Arnold et al. (2015) to train a DDPG-based actor that outputs a single query in the action representation space. $k$ nearest neighbor queries are retrieved and given to the

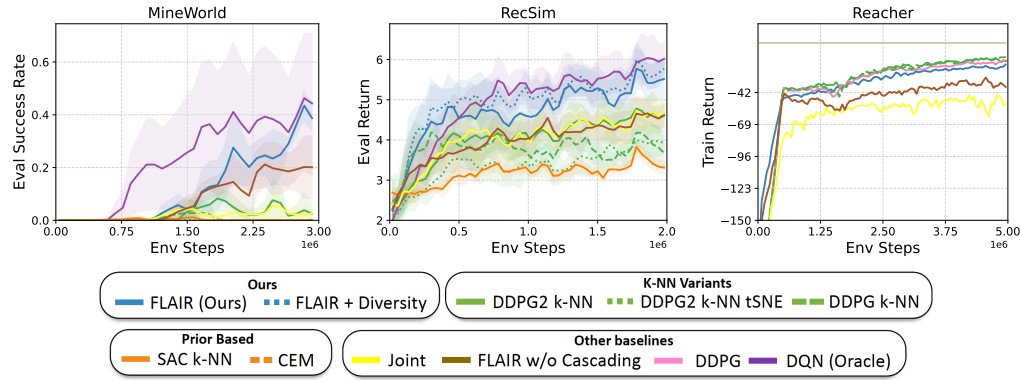

Figure 4: We evaluate FLAIR against baselines on the complex action space. The results are averaged over 3 - 6 random seeds, and the seed variance is shown with shading.

selection critic. Like the original paper, the actor is trained with gradients from the selection Q network. We adapt this method for continuous action spaces by sampling $k$ actions in a Gaussian centered at the selected action query.

- **DDPG2 k-NN**: In trying to make DDPG k-NN more robust, we include a stronger baseline that trains an extra retrieval critic network, only to train the retrieval actor. The selection Q-network is just used to act in the environment.

- **DDPG2 k-NN t-SNE**: One potential solution to a complex action space is to perform dimensionality reduction as a pre-training step over the action space. We apply t-SNE to the provided action representations and subsequently train DDPG2 k-NN. This method is not applicable to continuous actions.

- **CEM**: This baseline follows QT-Opt (Kalashnikov et al., 2018) in replacing the learned retrieval actor with a cross entropy method (CEM) (De Boer et al., 2005). Note that in terms of budget of evaluating the selection Q-function, CEM is very expensive as it requires multiple iterations of fitting a distribution over action samples. This method is only applied to the continuous action space environment.

- **SAC k-NN**: This method evaluates a distribution-based retrieval actor in the continuous action space, trained with SAC. The retrieved actions are sampled from the learned distribution, and the actual discrete actions are selected as the nearest neighbors of each list query.

- **DDPG**: This is the standard DDPG baseline for continuous action space tasks.

- **DQN (Oracle)**: We provide the result of DQN as the upper bound of our method on discrete action space tasks. It is an oracle because it has access to the entire list of actions as the retrieved samples.

- **FLAIR (Ours)**: We perform listwise action retrieval with a cascaded DDPG architecture.

- **Joint**: Instead of performing cascaded listwise action retrieval, an alternative is to train an actor with a joint action space of $k \times D$. Then, the action samples can be extracted from the joint action at intervals of $D$ dimensions each. This baseline is expected to be hard to learn because it does not utilize the fact that different dimensions are not completely independent, but rather form groups of actions.

- **FLAIR w/o Cascading**: We remove the list action encoder from the cascaded actors and critics, such that each actor-critic pair acts as an independent agent. By initializing the networks with unshared weights, these networks act as a multi-agent network without any communication. Thus, the cascaded actors are unaware of other actions in the list and cannot perform appropriate listwise RL. This ablation of our method demonstrates that FLAIR's listwise RL helps it improve task performance.

- **FLAIR + Diversity**: We augment our method with a reward to encourage diversity in the list actions for the retrieval agent only. This reward is not necessarily helpful as diversity in the retrieved list can sometimes be lead to interference in task learning.

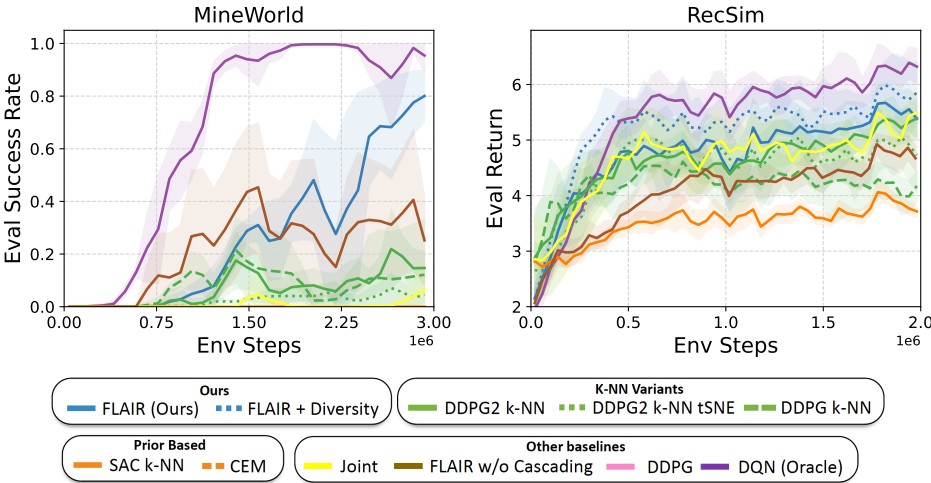

Figure 5: Consistency of our proposition across different action space conditions in terms of (Left) the action-dimension and (Right) the action-set size.

### 6.1.2 RESULTS

Figure 4 shows the complete experiment results.

- **Mine World**: We observe that FLAIR outperforms all the baselines, while FLAIR without Cascading is able to achieve some success as well. It is a complex action space as two different kinds of discrete actions available to the agent — navigation and mining actions. Furthermore, the stochastic noise added to the action representations makes the task even more complex. Therefore, because of the flexibility in the retrieval actor, FLAIR can learn well in this environment, and achieve close optimal performance as DQN (Oracle). Note that all methods and baselines learned on smaller versions of the maze with simpler action spaces.

- **RecSim**: FLAIR and its diversity variant outperform all the baselines and reach almost the sample efficiency of DQN (Oracle). The diversity reward helps FLAIR to explore in the beginning of training, but is not so helpful later. Joint, FLAIR w/o cascading are in the middle of the performance levels, while DDPG-kNN variants and SAC k-NN follow next. Crucially, we observe that DDPG2 k-NN outperforms DDPG k-NN, which shows the importance of learning an extra critic just for training the retrieval actor, unlike what was originally proposed in Dulac-Arnold et al. (2015).

- **Continuous Reacher**: FLAIR performs comparatively to DDPG, DDPG k-NN and DDPG2 k-NN approaches. Despite the non-linear matrix transformation, the resultant action space is still smooth enough for DDPG and variants to learn. DDPG k-NN and DDPG2 k-NN, by virtue of sampling 3 nearest neighbor actions to the continuous action, are more likely to optimize the selection Q-function than DDPG. Therefore, they stay above DDPG throughout the training procedure. Similarly, FLAIR and FLAIR w/o cascading learn faster in the initial stages of learning simply by having more candidates to perform directed exploration with. This demonstrates that listwise retrieval agent offers more flexibility than a local neighborhood approach for initial exploration by letting the agnet have more diverse action samples. CEM is unable to learn at all

### 6.2 ANALYSIS: CONSISTENCY ACROSS DIFFERENT COMPLEXITY OF ACTION SPACE

We evaluate our proposition against the baselines to examine the consistency of our result across different difficulty in the action space in the discrete RL tasks by changing the action-set sizes and the action-dimension.

**Action-set size**: Figure 5 (Right) shows the result of FLAIR compared to the baselines in the different action-set size from the main experiment. Empirically, we observed that the action set sizes largely influenced the sample efficiency of the learning. So here, we show the representative result with the action-set size equal to 5000 in RecSim simulator. Compared to the main experiment 4

which uses 10000 actions, all agents improved the performance(above the level of 5) as expected since the action-set size contributes to the complexity of the action space. Also, we observed that the margin between FLAIR and the baselines of Joint and DDPG2 k-NN became smaller, which indicates that our agent was able to learn efficiently in the larger and more difficult setup.

**Action-set dimension**: Figure 5 (Left) shows the result of FLAIR compared to the baselines in the different action dimension. In order to highlight the effect of the change in the action dimension, here we used 30 for the action dimension with the action-set size being 10 whereas in the main experiment 4 we used the dimension of 4 and the 100-tools. As we observed in the action-set size experiment above, all agents improved the performance due to the smaller set size. The result shows that the non-listwise method of FLAIR w/o Cascading deteriorated whereas our method was able to improve the performance.

## 6.3 Design Choices

We analyze the design choices of list encoder and weight initialization of cascaded actor-critic networks in Fig. 6.

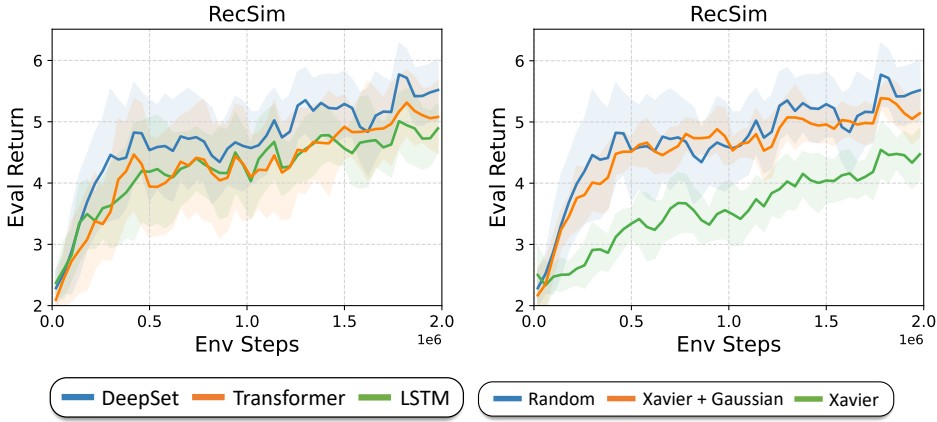

Figure 6: Validating the design choices of our method. (Left) We observe that DeepSet list encoder slightly outperforms Transformer and LSTM variants on RecSim. (Right) We observe that having more diverse weight initializations for different cascaded actor and critic networks enables a better performance of FLAIR.

### 6.3.1 Choice of List Encoder

We compare the performance of FLAIR for difference choices of list encoder architecture on RecSim environment. We observe that Deep Set slightly outperforms transformer (Vaswani et al., 2017) and LSTM list encoders. Because of its simplicity, we use deep set architecture for all our experiments.

### 6.3.2 Choice of weight initialization

We experiment with different ways to initialize the weights of cascaded actor and critic networks to see if having more diverse network initialization can lead to improved learning of listwise RL behavior. We observe that the standard Xavier initialization was outperformed by Gaussian and Xavier+Gaussian weight initialization. We believe the reasoning for this behavior is an encouragement of different actors to learn to act differently and focus more on the currently built list action.

## 7 Conclusion

We present FLAIR, a listwise action retrieval agent for performing flexible reinforcement learning in complex innumerable action spaces. We demonstrate its applicability to improve exploration initially and flexibility, generally, in two discrete action space RL environments, RecSim and Mine World, and one continuous action space Reacher environment.

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

APPENDIX

# A  ENVIRONMENT DETAILS

## A.1  MININGENV

The grid world environment, introduced in Sec. 5.1, requires an agent to reach a goal by navigating a 2D maze as soon as possible while breaking the mines blocking the way.

**State**: The state space is an 8x8-grid world, and each cell has three dimensions: (cell-type, mine-type-number, direction-number). When the corresponding type is matched, the second/third dimension would be meaningful. Otherwise, they would be set to be the default value as 0. To be more specific, there are four kinds of cells: empty, mine, goal, and agent:

1. Empty: each dimension equals 0

2. Mine: The cell-type dimension equals 1, the mine-type-number dimension equals the mine type id and the direction-number dimension equals 0. The mine set size is equal to 20 and it is divided equally into two parts: basic mines and complex mines. There is no significant difference between basic and complex mines except mine id. More specifically, the basic mine id range is [0, ..., 9], and the complex mine id range is [10, 19].

3. Goal: The cell-type dimension equals 2, and all the others equal to 0.

4. Agent: The cell-type dimension equals 3, the mine-type-number dimension equals 0, and the direction-number dimension equals to the corresponding direction number. There will only be 4 kinds of directions: facing up, down, left, right.

Ultimately, we will normalize each dimension to [0, 1] with each dimension's minimum/maximum value. And each time when resetting the environment, the category of each mine will be randomly changed.

**Termination**: An episode is terminated in success when the agent reaches the goal or after a total of 100 timesteps.

**Actions**: The base action set combines two kinds of actions: navigation actions and pick-axe(tool) actions.
The navigation action set is a fixed set, which contains four independent actions: going up, down, left, and right, corresponding with the direction of the agent. They will change the agent's direction first and then try to make the agent take one step forward. Note that, different from the empty cell, the agent cannot step onto the mine, which means that if the agent is trying to take a step towards a mine or the border of the world, then the agent will stay in the same location while the direction will still be changed. Otherwise, the agent can step onto that cell. An agent will succeed if it reaches the goal position.
The size of the pick-axe action set is equal to 100. Each tool has a one-to-one mapping, which means they can and only can be successfully applied to one kind of mine, and either transform that kind of mine into another type of mine or directly break it.
To state it more clearly, the toolset consists of basic and complex tools. Basic tools can only be applied to the basic mines, and will directly break the mine into an empty cell. As for the complex tools, they could only be applied to the complex mines, and some complex tools can turn the mines into one kind of basic mine. Others can directly break the mines.
Note, basic tools can't turn the basic mines into any other kind of mines, and also impossible for complex tools to turn the complex mines into another kind of complex mines. In addition, it is guaranteed that all mines can be broken, with one or two tools sequentially.

**Reward**: The agent receives a large goal reward for reaching the goal. The goal reward is discounted based on the number of action steps taken to reach that location, thus rewarding shorter paths. To further encourage the agent to reach the goal, a small exploration reward is added whenever the agent gets closer to the goal, and a negative equal penalty is added whenever the agent gets further to the goal. And also, when the agent successfully applies a tool, it will gain a small reward. When

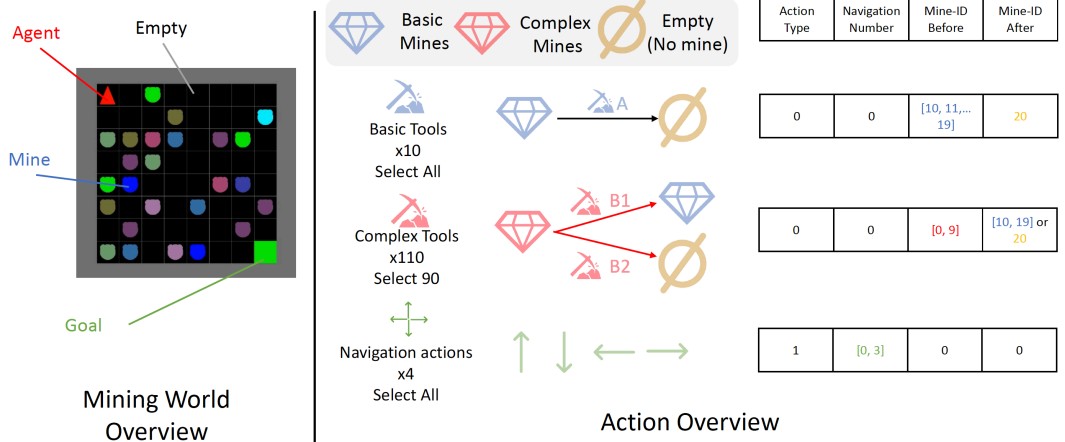

Figure 7: Mining Env Setting Description

the agent successfully breaks a mine, it will also gain a small bonus.

$$R(s,a) = \mathbb{1}_{Goal} \cdot R_{\text{Goal}} \left( 1 - \lambda_{\text{Goal}} \frac{N_{\text{current steps}}}{N_{\text{max steps}}} \right) +$$
$$R_{\text{Step}} \left( D_{\text{distance before}} - D_{\text{distance after}} \right) + \qquad (1)$$
$$\mathbb{1}_{correct\ tool\ applied} \cdot R_{\text{Tool}} +$$
$$\mathbb{1}_{successfully\ break\ mine} \cdot R_{\text{Bonus}}$$

where $R_{\text{Goal}} = 10$, $R_{\text{Step}} = 0.1$, $R_{\text{Tool}} = 0.1$, $R_{\text{Bonus}} = 0.1$, $\lambda_{\text{Goal}} = 0.9$, $N_{\text{max steps}} = 100$

**Action Representations**: The action representations are 4-dimensional vectors manually defined using a mix of number ids, and each dim is scaled to [0, 1]. as shown in Graph 7. Dimensions 1 identify the category of skills (navigation, pick-axe), 2 distinguish movement skills (right, down, left, up), 3 is used to denote the mine on which this tool can be successfully applied, and 4 are showing the result of applying this tool. We will normalize the action embedding space to [0, 1] for each dimension.

## B    NETWORK ARCHITECTURES

### B.1    CASCADED ACTOR

The whole actor has a cascaded format and each cascaded actor will receive two pieces of information: the state observation and the action list generated by previous cascaded actors. For one single cascaded actor, given the concatenation of the input components above, an 4-layer MLP with ReLU will process this information and generate one action. And this action will be concatenated with the previous action list. After transformed by an optional action-list-encoder, together with the state information, they become the input of next cascaded actor's input. In the end, the action list will be processed with 1-NN to find the nearest discrete action. After this, this action list will be delivered to the selection Q-network.

### B.2    CASCADED CRITIC

The critic has a one-to-one mapping relationship the actor. The whole critic consists of a list of cascaded critics and each cascaded critic will receive three pieces of information: the state observation, the action list generated by previous cascaded actors, and the action provided by the corresponding cascaded actor. For one single cascaded actor, given the concatenation of the input components above, an 2-layer MLP with ReLU will process this information and generate the action's value. This value will be used to update itself as well as the actor with TD-error.

### B.3 LIST ENCODERS

*List Encoder:* An intermediate list that contains the currently selected actions in between each cascaded actors is passed on to a encoder to summarize the list-action information.

**Bi-LSTM**: The raw action representations of candidate actions are passed on to the 2-layer MLP followed by ReLU. Then, the output of the MLP is processed by a 2-layer bidirectional LSTM (Huang et al., 2015). Another 2-layer MLP follows this to create the action set summary to be used in the following cascaded actor.

**DeepSet**: Similar to the Bi-LSTM variant of the summarizer, we employed the 2-layer MLP with ReLU followed by the mean pooling over all the candidate actions to compress the information. Finally, the 2-layer MLP with ReLU provides the resultant action summary to the following cascaded actor.

**Transformer**: Similar to the Bi-LSTM variant of the summarizer, we employed the 2-layer MLP with ReLU before inputting the candidate actions into a self-attention and feed-forward network to summarize the information. Afterwards the summerization will be the part of the input of the following cascaded actor.

### B.4 SELECTION Q-NETWORK

As for selection Q-network, it will received a concatenated information of state and one action to evaluate the q-value. By comparing the the q-value of each action, the highest q-value action will be chosen as the final output action.

## C EXPERIMENT DETAILS

### C.1 IMPLEMENTATION DETAILS

We used PyTorch (Paszke et al., 2019) for our implementation, and the experiments were primarily conducted on workstations with either NVIDIA GeForce RTX 2080 Ti, P40, or V32 GPUs on. Each experiment seed takes about 4 hours for Mine World, 8 hours for Reacher, and 6 hours for RecSim, to converge. We use the Weights & Biases tool (Biewald, 2020) for plotting and logging experiments. All the environments were developed using the OpenAI Gym Wrapper (Brockman et al., 2016). We use the Adam optimizer (Kingma & Ba, 2014) throughout.

### C.2 HYPERPARAMETERS

The environment-specific and RL algorithm hyperparameters are described in Table 1.

### C.3 HYPERPARAMETER TUNING

We found our methods to be sensitive to the following hyperparameters and we sweeped the associated values: actor-lr {0.01, 0.001, 0.0001}, critic-lr {0.01, 0.001, 0.0001}, actor decay {100K, 200K, 500K, 1M, 2M, 3M }, critic decay {100K, 200K, 500K, 1M, 2M, 3M}. Different environment settings need different combination of the hyper parameters.

| Hyperparameter | Mine World | Reacher | RecSim |
|---|---|---|---|
| **Environment** | | | |
| max episode steps | 100 | 250 | 20 |
| number of epochs | 1500/2500 | 8000 | 2000 |
| number of environments | 5 | 10 | 16 |
| action dim | 30/4 | 5 | 30/60 |
| number of all actions | 14/104 | - | 10000 |
| obs encoder applied | True | False | False |
| state(obs) dim | 3 | 11 | 30/60 |
| **RL Training** | | | |
| lr | 0.001 | 0.001 | 0.001 |
| WOLP actor lr | 0.001 | 0.001 | 0.0001 |
| WOLP critic lr | 0.001 | 0.001 | 0.001 |
| Q-network hidden dimension | 128_128 | 400_300 | 64_32 |
| candidate list len | 4 | 3 | 4 |
| batch size | 256 | 256 | 256 |
| buffer size | 500000 | 1000000 | 1000000 |

Table 1: Environment/Policy-specific Hyperparameters

