# OpenReview forum: "Hedge Your Actions: Flexible Reinforcement Learning for Complex Action Spaces"
_ICLR.cc/2023/Conference — Submitted to ICLR 2023_

### Official Review · Reviewer_ieEx · 2022-10-23

**Confidence:** 4
**Correctness:** 2
**Technical Novelty And Significance:** 2
**Empirical Novelty And Significance:** 1
**Recommendation:** 3

**Clarity, Quality, Novelty And Reproducibility:**

The paper has extremely limited novelty, it's fairly clearly written but completely lacks reproducibility. There is not a single Equation in this paper, not a single loss function defined or analyzed.

**Strength And Weaknesses:**

Weaknesses:

As i understood, the paper proposes FLAIR, according to the authors a novel method for performing action retrieval in complex action spaces. Indeed the authors claim in their main contribution is introducing the problem of complex innumerable action spaces in reinforcement learning. Then they solve this task by proposing FLAIR. This claim is clearly not true, because right after making the claim of introducing this setting, the authors continue with a related work section discussing other methods that solve this task, most notably Cascaded DQN (CDQN) the base method from which FLAIR is derived.

The authors makes unsupported claims throughout the  paper like:
"FLAIR make robust actions retrieval" - Robust in what sense? is it guaranteed to include always the optimal action? Is it guaranteed to retrieve in high probability? Is robust is defined in the paper?
"FLAIR enables directed exploration of the task" - How does it do this? Directed exploration has a clear definition in RL and an immense body of work both in discrete and in continuous action spaces. The authors repeat this claim several time in the paper but never offer any intuition as to why, or any empirical evidence in support of this.

-- Reproducibility is a main issue of this paper---
The paper does not contain a single equation. No loss functions are defined, for none of the K critics or K actors described in the methodology section. How is a reader supposed to reproduce the presented results when no detail on even the loss functions is given?

Novelty is also a weakness. In my opinion the paper is just a modification of Cascaded DQN with DDPG as the learning algorithm. I cannot even completely verify this, as no pseudocode of the method was provide, no loss functions were defined and no source code was given.
Now this would be fine, if clear empirical performance difference were observed and explained between the two versions, or if some intuition was given as to why DQN is not suitable, except for the fact that it cannot be applied to continuous actions.

The empirical evaluation is also extremely lacking. The method makes not technical or theoretical contributions, so the bar for empirical contributions is increased. The authors use 3 benchmarks, 2 with discrete actions and one continuous actions. The choice of baselines to compare is also questionable. The authors modified CDQN to recover FLAIR and make the choice not to compare the two algorithms, so we cannot even verify if replacing DQN with DDQN has any empirical effect. Moreover, the authors include the performance of DQN as an oracle, claiming that their method couldn't possibly reach the performance of DQN since they do not have the full action space and can only pick a subset of the actions. And this is clearly the main issue with the experimental evaluation. The whole point of the setting is that methods that directly work in discrete actions spaces like DQN would fail here because the number of actions is so large. DQN does have access to the complete action space, but the authors should use benchmarks where this hinders learning. Showing benchmarks where DQN already achieves good performance clearly does not make FLAIR or any action retrieval method for RL needed.

**Summary Of The Paper:**

The paper proposes a method to solve large action space control tasks via a two step RL procedure. The authors first perform action retrieval over a fixed space of action representations and then perform a selection procedure over the retrieved actions via a trained value-function. The proposed method is evaluated over a benchmark of 3 discrete and continuous tasks, including a simulator of a recommender system.

**Summary Of The Review:**

This paper clearly is not ready for publication, especially in a venue like ICLR.
First the method in provides limited novelty, as it is a modification of a previously published algorithm, with CDQN (cascaded DQN) by using DDPG as base learner instead of DQN. While this would be fine if the method clearly improved performance over the original algorithm, or provided some theoretical/intuition over the importance of this modification, the authors fail to do any of that. Moreover, The experimental evaluation also is extremely lacking as discussed in the section on Weaknesses.

---

> ### Author Response · Authors · 2022-11-19
> **Addressing concerns on Novelty, Technical Details, and Evaluation (Qualitative and Quantitative)**
>
> We thank the reviewer for extensive and detailed feedback. While the reviewer’s critiques are justified, we respectfully believe the major issues are largely due to some misunderstandings - which have surely arisen due to missing details in our paper writing (about CDQN and FLAIR), despite our best attempts.
>
> ### FLAIR vs Cascaded DQN
> **Novelty of Problem statement** + **Why is CDQN not for our baseline?**
>
> We acknowledge that the innumerable action space is an existing problem (Sec 2.1) and did not intend to claim this as a new problem for our contribution. Instead, we identify a specific problem where prior innumerable action space approaches fail to retrieve actions well: the challenge of action space complexity (Edited Sec 3.3 for more clarity).
> - **CDQN does not work for innumerable action spaces**: CDQN is a value-based method (e.g., DQN) proposed for listwise RL tasks in discrete action spaces (such as recommending a coherent list of 5 interesting videos on social media). So, it enumerates all actions just like DQN and is not applicable to innumerable action spaces (Please check new experiments in Fig 5 about how DQN fails in large action spaces). In particular, when the cascading list size of CDQN is 1 it’s just DQN. Thus, CDQN is not needed as an extra baseline other than DQN, since our final task does not require list generation. Thanks to your insight, we have augmented Sec 2.1 for better clarity.
> - **CDQN does not work for continuous action space listwise action retrieval**: All prior listwise RL works are in discrete action space, whereas we need to perform listwise action retrieval in the continuous space of action representations. While we borrow the idea of “cascading” networks to enable listwise communication between agents, the exact implementation of FLAIR is significantly different because of (a) using action representations, (b) continuous action space, (c) having listwise encoder, and (d) learning paired actor-critic architecture for each list index. We have added these details in Sec 4 (Approach) to clearly demonstrate these differences.
>
> ### Missing technical details of FLAIR + Technical details for Reproducibility
> We agree that insufficient explanation on FLAIR in the original writing could cause the confusion on the novelty of our method compared to CDQN and harm the reproducibility. Thus, in Sec.4, we improved the clarity and added the technical details of FLAIR, including exact mathematical formulation and loss functions(e.g., TD-error). We further add Algorithm for our method in Appendix A, provide code for all experiments (including rebuttal), and network and hyperparameter details in Appendix.
>
> ### Improving Empirical Evaluation
> - **Demonstrate Innumerable action space is a challenge for DQN (Oracle)**: We agree it is important to demonstrate why conventional value-based approaches like DQN do not scale to innumerable action spaces. We demonstrate this with new experiments in Fig.5 and Appendix C.2, where we analyze the memory inefficiency of Oracle DQN w.r.t FLAIR. We vary two factors to enlarge action spaces, namely Dimensionality ([16, 32, 48, 64]) and Action-set size ([5k, 10k, 50k, 100k, 150k]) on RecSim environment. DQN goes Out-of-Memory at 100k actions, while FLAIR is only at 20% GPU usage. Since the selection network in FLAIR only evaluates a small number of candidate actions (3 for RecSim), its memory requirements are much lower than DQN where Q-network enumerates and evaluates all actions. Thus, retrieval-selection frameworks are essential in practical applications where the action space grows rapidly and all the neural networks are at a large practical scale (e.g. due to larger state spaces). We also show that FLAIR’s suboptimality against DQN does not increase with complexity. Note that, we keep the main results in settings where DQN could still work, so that we can estimate the best-case oracle performance of the retrieval-selection methods.
> - **Extensive Quantitative Results**: Firstly, in Fig.4 we ensured the full convergence of all algorithms for a fair performance comparison to FLAIR. Also, in Sec.6.4 and C.1, we conducted further experiments where FLAIR consistently outperorms DDPG2 k-NN (the strong baseline in Fig.4) across different complexities of action spaces to demonstrate FLAIR consistently outperforms the baselines. Finally, we added more experiments to justify our design choices on the cascading architecture of FLAIR in Figure 7 and C.3.
> - **Extensive Qualitative Results**: We added the qualitative analysis in Fig.6a to qualitatively analyze the properties of actions retrieved by FLAIR (which learns to output a list of actions) and DDPG2 k-NN (outputs a k-neighborhood of actions). In short, we found that cascading actors were able to provide the candidate actions consisting of multiple optimal actions that helps the selection network’s exploration in task.

---

> > ### Comment · Reviewer_ieEx · 2022-11-24
> > **Response to rebuttal**
> >
> > I thank the authors for their reply to my concerns.
> > After some considerations, I have decided to increase my score, but nevertheless I still think the paper is not ready for publication. I will list my reasons below:
> > - The paper in it's current form, while in a substantially better state than the initial submission, comes with substantial modifications. The authors have highlighted in blue what they claim are the changes they made to the paper, but after running pdfdiff to the two submissions, around 80% of the lines of the paper have been modified. Keeping apart from the fact that the authors only highlighted in blue a part of the changes and not all of them, judging the paper now would require to do a completely new review from the beginning which in my view it is not the objective of the rebuttal phase.
> > - While the experimental section has been substantially improved, and I understand the authors argument that in large scale experiments DQN and CDQN will run out of memory, I still do not understand the point then of the experiments shown in Fig. 4. In these experiments, DQN does not run out of memory, so CDQN should also be compared since it is the method from which FLAIR takes inspiration. Moreover, again calling DQN an oracle here is still misleading. DQN is by no means an oracle, calling it an oracle and claiming that it is an upper bound of the performance of action retriaval methods is clearly not true. DQN by no means is guaranteed to converge to the optimal policy, even in the case of 2 actions, let alone in the setting presented here.
> > - The paper continues to make unsupported claims. The main claim at the end of Section 1, which the authors accepted was not true in their rebuttal to my review is still present in the modified paper. "Our primary contribution is introducing the problem of complex innumerable action spaces in reinforcement learning." The authors clearly did not introduce this setting, but continue to claim that it is the primary contribution of the paper also after the rebuttal. Similar comments still apply to the claims about "robustness" and "exploration capabilities" of the method.

---

> > > ### Author Response · Authors · 2022-11-25
> > > **Thank you. We attempt to clarify the misinterpretations.**
> > >
> > > We thank the reviewer for taking the time to respond and providing a clear list of their leftover concerns. We believe these are still because of a misunderstanding of our paper’s ideas and prior work. We would like to attempt to clarify this again.
> > >
> > > &nbsp;
> > >
> > > ### 1. **Complex** and **Innumerable** are different.
> > > - Innumerable = *large* discrete OR continuous action space => infeasible to enumerate all actions and run a Q neural network on all of them. (Sec 3.1)
> > > - Complex = *Unsmooth* action space => when actions with nearby representations (e.g. $a$ and $a + \Delta a$) have different effects on the task. (Sec 3.3)
> > >
> > > **Innumerable** action space is not a new problem setup — and was **never claimed** to be (see our intro, related work, and problem formulation). However, **Complex AND Innumerable** action space is our new problem setup. Here, prior innumerable approaches suffer (baselines in Sec 6.1).
> > >
> > > Thus, we respectfully maintain that our primary contribution is “introducing the problem of **complex innumerable** action spaces in RL”.
> > >
> > > &nbsp;
> > >
> > > ### 2. **CDQN = DQN**. Already in Experiments.
> > > CDQN [1] stands for Cascading DQN. It is simply a sequenced array of multiple DQNs. CDQN reduces to DQN when array length = 1. To elaborate, consider the following examples of RL problem families:
> > > - (1) **Discrete action-space RL**: methods like DQN select ONE discrete action out of many (e.g. going Up or Down in Pong).
> > > - (2) **Continuous action-space RL**: methods like DDPG and SAC output an N-dimensional continuous action.
> > > - (3) **Listwise Discrete action-space RL**: a special class of tasks involving selecting $C$ discrete actions out of many actions. E.g. recommend a **list** of $C=5$ videos on YouTube. Methods like CDQN are used here, having $C=5$ DQNs in sequence.
> > >
> > > Now, our environments (MineWorld and RecSim) are standard discrete action-space RL (Family 1), i.e. select ONE tool in MineWorld or select ONE item to recommend in RecSim. Applying CDQN here means $C=1$. **But this is exactly DQN**, which is already there in experiments.
> > >
> > > Finally, there is a **novel** family of RL we introduce in this paper:
> > > - (4) **Listwise continuous action space RL** — We utilize this to retrieve a *list of continuous action* representations. Our proposed method FLAIR is the first to address this setup.
> > >
> > > &nbsp;
> > >
> > > ### 3. Why is DQN an oracle?
> > > An oracle is a solution that uses privileged access to solve a problem [2]. DQN is an oracle as it utilizes **privileged compute resources**. To elaborate,
> > > - In retrieval-selection frameworks (Fig. 1), the selection network is a DQN. Finding the best action on DQN requires finding an arg max over all N actions in the action space.
> > > - But with a retrieval agent, we retrieve L actions only (L << N), so that we can still approximate arg max with few actions, such as 3.
> > > - Thus, all retrieval-selection networks utilize a much smaller GPU memory, as DQN is now evaluated on L actions instead of N.
> > > - This makes all retrieval-selection frameworks (Sec 2.1) an approximation to DQN, as they can never do a true arg max like DQN.
> > > - Thus, DQN uses privileged GPU memory as an oracle.
> > >
> > > We believe reporting results on DQN (i.e. no retrieval) in Fig 4 is important to understand what is the ceiling of performance of all retrieval-selection methods in **complex action spaces**. At the same time, Fig 5 added on your suggestion is extremely valuable, to demonstrate exactly when DQN fails to scale and how our method still performs and scales well.
> > >
> > > &nbsp;
> > >
> > > ### 4. “Robustness” and “Exploration”
> > > Maybe you missed Part 2 of our response?
> > > - Robustness: We say our method is **“robust to complex action space”**. All this means is that our method works well in various complex action spaces (verified in experiment sections 6.1 - 6.4). In response to your review, we had made sure that every mention of robustness in the paper is accompanied by this clarification.
> > > - Exploration: Our method improves exploration by being able to **retrieve various different actions** (already explained in Introduction and justified by quantitative and qualitative experiments). In our revision, we already made sure to remove the term "directed" (exploration) from the paper based on your recommendation.
> > >
> > > &nbsp;
> > >
> > > ### [References]
> > > - [1] Chen, Xinshi, et al. "Generative adversarial user model for reinforcement learning based recommendation system." ICML. PMLR, 2019.
> > > - [2] Baselines and Oracles: https://ofirnachum.github.io/posts/baselines-and-oracles/
> > >
> > > &nbsp;
> > >
> > > While we still maintain that the major issues are due to misunderstandings, they must be due to shortcomings in our original paper writing. This is exactly why we made every effort to improve clarity in our revised paper by editing and reordering the writing (which is why it appears different in pdfdiff). Again, we appreciate your valuable feedback which has improved the paper significantly. We understand if you decide to not update your score, but at least **hope that all the concerns are addressed**.

---

> > > ### Author Response · Authors · 2022-12-05
> > > **Response reminder, thank you.**
> > >
> > > Hello Reviewer ieEx, this is a gentle reminder that we have responded to your most recent reply. Thanks to your latest comments, we answer:
> > > 1. **Complex and innumerable are different aspects**: innumerable action space is well-known, while complex + innumerable problem is our novel contribution.
> > > 2. **CDQN is already in experiments**: CDQN=DQN when the environment requires one action. Fig. 4 shows how DQN (and CDQN) don’t scale to a large action space. FLAIR is a solution to a novel setup of listwise continuous action space RL and only borrows the idea of using multiple Q-networks from CDQN.
> > > 3. **Why DQN is an oracle**: it uses privileged access to GPU memory to compute true arg-max on all actions. FLAIR and all other baselines approximate arg-max by selecting high-quality actions, without using high GPU memory (which bottlenecks quickly for DQN in Fig. 4).
> > > 4. **Robustness** is defined with respect to complex action space. **Exploration** is attributed to the ability to retrieve more than one action. (Part 2 of our rebuttal)
> > >
> > > Thanks again for your valuable time. Your constructive feedback, detailed questions, and comments have helped us better clarify the connection of FLAIR to baselines (e.g., DQN and CDQN). We hope we have addressed all of your concerns, and if not, please let us know how we can clarify anything else.

---

> ### Author Response · Authors · 2022-11-19
> **Addressing Writing and Reproducibility Concerns (Part 2 of 2)**
>
> ### Writing
> **Missing definitions: *“Robust”* action retrieval**: We meant “robustness” in the context of FLAIR being robust to complexity in innumerable action spaces. Thus, we call FLAIR’s action retrieval “robust” as it’s providing flexibility in the actions it can sample as compared to baselines. Based on your recommendation, we have edited the text throughout to clarify this.
> **Missing definitions: “Directed exploration”**: We have clarified and elaborated the meaning of exploration that listwise RL improves. In FLAIR, the selection network stands on the candidate-set that has been chosen to maximize the environment reward by the cascaded actors. Thus, the exploration on this candidate-set enables is influenced by the action retrieval, this we call it as the directed exploration in FLAIR. In order to analyze the properties of the retrieved list, we also added the qualitative result in Fig. 5 showing that FLAIR was able to maintain the multiple solutions in the candidate-set helping the selection network explore towards the reward maximizing direction.
>
> ### Reproducibility
> Unfortunately, earlier the code submission was not possible since the forum was not open. So we have attached the codebase with full details to reproduce all the results in README to improve the reproducibility. Also, from a writing perspective, we hope that improved clarity on the technical details of FLAIR in Sec.4 and the description of important protocols in FLAIR in Algorithm1 in Appendix A would help improve the reproducibility.

---

### Official Review · Reviewer_GAf6 · 2022-10-25

**Confidence:** 3
**Correctness:** 3
**Technical Novelty And Significance:** 3
**Empirical Novelty And Significance:** 2
**Recommendation:** 3

**Clarity, Quality, Novelty And Reproducibility:**

Clarity:
- The paper is clearly written and motivates the problem very well.

Quality:
- The paper lacks proper explanation of the method and the evaluation is also not sufficient.

Novelty:
- The proposed method is novel to the extend of a new method in retrieving actions. The method is original to some extent.

Reproducibility:
- It's difficult to see if the results are reproducible as sufficient information is not provided. Appendix has implementation details but the method is not clear in the main text.

**Strength And Weaknesses:**

Strengths:
- The paper motivates the problem very well. The abstract and introduction are highly aligned with the problem targeted.
- The paper uses three simulation environments for training and evaluation of the proposed method and several baselines are used for performance comparison.

Weakness:
There are several weakness on writing, explanation, and evaluation. The justification is below:
1. Results:
- The paper motivates the problem very well but fails to justify using proper experiments. For example, in section 6.1.2, authors claim that their proposed is better than all baselines. However, it is not seen in neither Figure 4 nor Figure 5 and DQN (oracle) seems to be performing best among all. The y-axis of both the figures is eval success rate or eval return means higher the AUC better the algorithm. This is not seen in any of the figure.
- The results are not extensive and hence it is difficult to conclude that the method is better. Only Figure 4 & 5 are given as performance comparison and no quantitative results are available. Figure 6 is for the representation learning and doesn't directly come under contributions.
- The retrieval selection framework is not very common so author should justify all steps of how retrieval works in their method. I couldn't find sufficient information on the same.

2. Method:
- It's difficult to find why the method would work best. The approach is explained in sec 4 which is not sufficient. For example, why would you be using cascaded actor-critic as compared to normal actor-critic. Not clear at all, why the design choices are used?
- The authors have cited several papers on which the proposed methods is based (for example, Deep Set, Cascaded DQN, listwise DQN, etc.). These citations make the paper highly abstract and not sufficient for the reader to understand the proposal. I would suggest to make the proposed method explanation in the main text itself in Sec 4.

3. Writing:
- The title is to hedge your actions. I have not found any useful information where hedging of actions is justified.
- Sec 3.2, framework 1 should be Figure 1.

**Summary Of The Paper:**

The paper proposed a new method for complex and large action space. The proposed method uses listwise RL for the retrieval task which is further used for action selection.

**Summary Of The Review:**

The paper has several weaknesses and it is not ready for publication. See weaknesses for justification.

---

> ### Author Response · Authors · 2022-11-19
> **Response (Part 1 of 2)**
>
> We thank you for your detailed and constructive feedback. We appreciate your positive comments on the motivation, environments, and baselines. We address your concerns about writing and evaluation with the following modifications and experiments:
>
> ### Experimental Evaluation
>
> #### DQN (oracle) v/s FLAIR in Large Action Space
> We agree with you that the results(Fig.4&5) conflicted with our claim of DQN being infeasible in the innumerable action space. So we performed the additional analysis illustrating the gravity of the problem setting in our experiments as follows;
> In Fig.6 and Appendix C.2, we analysed the memory efficiency and optimality of FLAIR compared to the discrete oracle(i.e., DQN) by varying the two factors to enlarge action spaces, namely Dimensionality ([16, 32, 48, 64]) and Action-set size ([5k, 10k, 50k, 100k, 150k]) on RecSim environment. Since the selection network in FLAIR only evaluates a small number of candidate actions (3 for RecSim), its memory requirements are much lower than DQN where Q-network enumerates all actions. Thus, retrieval-selection frameworks become essential in practical applications where the action space grows rapidly and all the neural networks are at a large practical scale (e.g. due to larger state spaces).
>
> #### Extensive results
> **Quantitative results**
> Firstly, in Fig.4 we ensured the full convergence of all algorithms for a fair performance comparison to FLAIR. Also, in Sec.6.4 and C.1, we conducted further experiments where FLAIR consistently outperorms DDPG2 k-NN (the strong baseline in Fig.4) across different complexities of action spaces to robustify superiority of FLAIR. Finally, we added more experiments to justify our design choices on the cascading architecture of FLAIR in C.3.
>
> **Qualitative results**
> The website contains qualitative results. (i) Mining: Baseline fails to select the corresponding digging tool because of a large number of tools and a noisy action space, (ii) Even in a simple and widely used RL environment like Reacher, a complex action space transformation makes all methods significantly suffer in sample efficiency. This demonstrates the importance of our problem statement. Also added a discussion to these in the paper.
> We added the qualitative analysis in Fig.5 to qualitatively analyze the properties of actions retrieved by FLAIR (which learns to output a list of actions) and DDPG2 k-NN (outputs a k-neighborhood of actions). In short, we found that cascading actors were able to provide the candidate actions consisting of multiple optimal actions that helps the selection network’s exploration in task.

---

> ### Author Response · Authors · 2022-11-19
> **Improvements to Writing (Part 2 of 2)**
>
> ### Writing
> **More details on Retrieval-Selection Framework**
> We added further technical details to Section 3.2 regarding Retrieval-Selection Framework. Taking the example of Dulac-Arnold et. al (2015), they employed a DDPG actor to provide a continuous query embedded in the action representation space as all other discrete actions. Then, they performed the k-nearest neighbor search to find the k nearest actions to the query based on its Euclidean distance. In the same spirit, Kalashnikov et al., (2018) utilized a cross entropy method to iteratively refine the distribution to sample queries from.
>
> **More details on Approach**
> We improved the clarity of Sec.4 to make the explanation of FLAIR self-sufficient. But due to the limited space, we added more discussions regarding the network architecture and the exact algorithm in Appendix.
>
> **Why would FLAIR work the best in Complex action spaces?**
> As discussed in Sec 3.3, complex action spaces cause the un-smoothness in action representation space that neighbouring actions in the space possibly have totally different utilities on the environment. Prior works including the normal actor-critic retrieve the actions assuming that neighbouring actions have similar utilities. Thus, the candidate-set would likely to be not ideal for the selection network to explore. To approach this, we posit that action retrieval can be most generally formulated as listwise RL, where the agent must output a list of actions suitable for the task of optimizing the selection Q-network and propose the simplest way to incorporate listwise RL in the action retrieval. Finally, to support our claim, we added the qualitative analysis (Fig.5) to qualitatively analyse the properties of the retrieved candidate-set.
>
> **Justification of Design Choices**
> We add Sec 6.5.1 experiments, where we validate our design choice on the cascaded-critics by comparing them to two variations; (i) *Single-Critic* that trains a shared extra critic evaluating cascading actors individually like DDPG2 k-NN, and (ii) *Joint-Critic* that trains a single joint critic evaluating cascading actors’ output as a whole at once. As a result, we confirmed that Cascaded-Critics of ours consistently outperforms the others in both MineWorld and RecSim environments.
>
> **Title**
> We agree with your helpful feedback and have edited out the term “hedge” from the title to improve clarity. You are right in saying that hedging or diversifying only happens occasionally and could be distracting from the core problem of complex innumerable action spaces.
>
> **Reproducibility**
> Unfortunately, earlier the code submission was not possible since the forum was not open. So we have attached the codebase with full details to reproduce all the results in README to improve the reproducibility. Also, from writing perspective, we believe that improved clarity on technical details of FLAIR in Sec.4 and the description of important protocols in FLAIR in Algorithm1 in Appendix A would help improve the reproducibility.

---

> ### Author Response · Authors · 2022-12-05
> **Response reminder, thank you.**
>
> Hello Reviewer GAf6, this is a gentle reminder that we have [responded](https://drive.google.com/file/d/1jKkLqk7-9o9bE6c6-l8XyWqlTGz9C3VL/view?usp=share_link) to your review. Thanks to your comments, we have added:
> 1. **Experiments**:
> a. DQN (Oracle) does not scale to large action spaces $\rightarrow$ Added Fig.5 and Appendix C.2.
> b. Extensive quantitative results $\rightarrow$ More results on different action space complexities and design choices (Sec 6.4).
> c. Extensive qualitative results $\rightarrow$ Fig.6 and the videos on the [website](https://sites.google.com/view/complexaction) analyzes the list of actions retrieved by FLAIR.
> 2. **Writing**
> a. Updated the text with more technical details on retrieval-selection frameworks (Sec 3.2) and approach (Sec.4).
> b. Why listwise RL of FLAIR works the best in Complex action spaces (Sec 3.3) $\rightarrow$ prior works assume smooth action space.
>
> Thanks again for your review comments, they have really helped us better emphasize why FLAIR outperforms baselines! We hope we have addressed all of your concerns, and if so, please consider raising your score.

---

### Official Review · Reviewer_oShM · 2022-10-25

**Confidence:** 4
**Correctness:** 3
**Technical Novelty And Significance:** 3
**Empirical Novelty And Significance:** 3
**Recommendation:** 6

**Clarity, Quality, Novelty And Reproducibility:**

- Clarity: The paper is generally well written and easy to read. There are some aspects where it could be improved:
  - ‘Hedge’ is only mentioned three times, two of which are in the title and abstract, where it is not explained; the third time, in the introduction, it appears to be a synonym for diversifying. Please consider using a different title, or strengthening the connection to the text. The main topic of the paper doesn't seem to be diversity/hedging, but rather dealing with innumerable action spaces in RL, so this title might not be ideal.
  - The second sentence of the abstract is not clear to me, possibly because I am not sufficiently familiar with the recommender systems literature. However, presuming that I am part of the target audience, please clarify why the representations of the items in a recommender system don’t apply to all users in the same way.
  - The acronym ARDDPG is not introduced, please fix (is it Action Retrieval DDPG?).
  - The caption of figure 5 does not make clear how the experiments presented there differ from those in figure 4 (action set size
- Quality: See strengths and weaknesses.
  - Writing quality: please language edit sections 2.1-2.3
- Novelty: the paper extends the explicit treatment of complex action spaces through cascading DQN from Chen et al. to continuous action spaces, and makes the case for introducing the list-wise RL approach to the wider RL literature.
- Reproducibility: no concerns.

**Strength And Weaknesses:**

- Strength: the proposed method looks like it has wide applicability.
- Strength: the paper contains extensive baselines and ablations.
- Strength: the paper addresses a relevant problem.
- Weakness: the learning curves shown in figures 4 and 5 don’t seem to have converged, and don’t seem overly long (a few million environment steps), leaving open the question of longer-term performance.
- Weakness: There are no experiments where the ‘oracle’ DQN is not a feasible solution (it is presented for all experiments, and outperforms all other baselines and the proposed method). There is also no analysis to show where that would happen, leaving open the question how necessary the proposed method is. More discussion of required network sizes and sample efficiency (and hence possibly also compute complexity) would be welcome.


**Summary Of The Paper:**

The paper proposes a method to deal with complex ('innumerable') action spaces in RL. The method is based on one from the recommender system literature, cascaded DQN, where a list of action candidates is produced by an internal actor trained through RL, from which the final one is selected. The proposed method is tested on a few different domains, where it is shown to outperform all chosen baselines except a DQN, which is considered the 'oracle'.

**Summary Of The Review:**

The paper presents an interesting idea, explained clearly and placed well in the context of the existing literature. The experimental evaluation is strong on some points (baselines, ablations), but does not fully manage to convince due to the relatively short nature of the training runs. Furthermore the gravity of the problem (and hence the necessity of the proposed solution) is not supported numerically. Hence, the paper leaves open some important questions. If those could be addressed, I would increase my score.

---

> ### Author Response · Authors · 2022-11-19
> **Response to Review**
>
>
> We thank you for your elaborate constructive feedback. We appreciate your positive comments on the method, application, baselines, and ablations. We address your evaluation concerns with the following experiments and changes:
>
> ### Learning might not have converged in experiments
> While we ran all the methods on all environments for longer, we had originally chosen the stopping point ensuring all the methods have converged. However, we agree with your insight that for a reader this might leave questions. Thus, we have replaced all curves with longer environment steps in Figure 4, clearly demonstrating convergence. The trends are consistent as before.
>
> ### Numerically justifying the gravity of the problem: Experiments where DQN (Oracle) fails
> We agree it is important to demonstrate why conventional value-based approaches like DQN do not scale to innumerable action spaces. We demonstrate this with new experiments in Fig.5 and Appendix C.2, where we analyze the memory inefficiency of Oracle DQN w.r.t FLAIR. We vary two factors to enlarge action spaces, namely Dimensionality ([16, 32, 48, 64]) and Action-set size ([5k, 10k, 50k, 100k, 150k]) on RecSim environment. DQN goes Out-of-Memory at 100k actions, while FLAIR is only at 20% GPU usage. Since the selection network in FLAIR only evaluates a small number of candidate actions (3 for RecSim), its memory requirements are much lower than DQN where Q-network enumerates and evaluates all actions. Thus, retrieval-selection frameworks are essential in practical applications where the action space grows rapidly and all the neural networks are at a large practical scale (e.g. due to larger state spaces). We also show that FLAIR’s suboptimality against DQN does not increase with complexity. Note that, we keep the main results in settings where DQN could still work, so that we can estimate the best-case oracle performance of the retrieval-selection methods.
>
> ###  More discussion of required network sizes and sample efficiency
> We added the discussion of the important design choices (e.g., Architecture of critic and List-encoder) in Approach (Sec 4) and experiments (Sec.6.5). We additionally added more detailed discussions regarding the algorithm in Appendix A, and architectural choices of our contribution of listwise retrieval in Appendix C.3. Finally, we improved the clarity on the discussion of the tuning of common yet important hyper-parameters across baselines in Appendix E.3.
>
> ### Writing Clarity
> - **"Hedge" in the title**: We agree with your (and Reviewer GAf6’s) helpful feedback and have edited out the term “hedge” from the title to improve clarity and instead maintain focus on the core problem of complex innumerable action spaces.
> - **RecSys example in abstract**: Thank you for providing your perspective, we agree that the writing can be clarified for a general audience, and have edited accordingly. To answer your question here, action representations in a movie recsys are often pretrained on generic features like genre and ratings. However, user preferences are diverse (e.g., depending on their age) and vary over time (e.g., depending on current mood). Thus, two nearby movies in this generic space could have vastly different values for a particular user.

---

> ### Author Response · Authors · 2022-12-05
> **Response reminder, thank you.**
>
> Hello Reviewer oShM, this is a gentle reminder that we have [responded](https://drive.google.com/file/d/1jKkLqk7-9o9bE6c6-l8XyWqlTGz9C3VL/view?usp=share_link) to your review. Thanks to your comments, we have added:
> 1. **Experiments**: Updated fully converged curves (Fig.4) and justify the gravity of problem by showing DQN does not scale to large action spaces (Fig.5 and Appendix C.2),
> 2. **Writing Changes**: Added important design choice discussion (Sec 4) and experiments (Sec 6.5), e.g., architecture of critic and List-encoder. Added algorithm in Appendix A, updated title and abstract.
>
> Thanks again for your time and review comments, they have really helped us better illustrate the importance of FLAIR. We hope we have addressed all of your concerns, and if so, please consider raising your score.

---

### Official Review · Reviewer_CZhi · 2022-10-31

**Confidence:** 5
**Correctness:** 2
**Technical Novelty And Significance:** 3
**Empirical Novelty And Significance:** 3
**Recommendation:** 8

**Clarity, Quality, Novelty And Reproducibility:**

The approach is novel and is intriguing.  The work appears reproducible and uses accessible test domains.  The approach is explained well.  The authors include implementation level details.

**Strength And Weaknesses:**

I think the stacked actors is a great way to produce a list of candidate actions and selecting a winner using a final Q-function selection network is a solid approach.  The paper does a good job of justifying this approach.

The graphic in Figure 2 does a good job of showing the flow of the approach.

I was surprised that the upstream actor outputs are fed into the critic along with the corresponding actor.  It would have been nice to see some justification of this design decision: is this needed and does it help?  Ensembles of critics (or Q-functions) can be pretty diverse and unstable and, I'd think, reliable in occupying different parts of solution space to produce diverse learning signals.  I also didn't understand how these other actions affect the critic output or how this is driving the ensemble to have more diversity (assuming this is the purpose).

In my view, the biggest weakness is that this work left one assumption unstated and unexamined.  This assumes that the selection network has the ability to represent the potentially volatile changes in Q-value for nearby action for a given state.  Is there any work showing that the function modeled by an actor is more smooth than the function represented by the critic?  This work didn't provide any analysis that the selection network was capable of representing substantial changes in Q-value from nearby actions.  As I understand this work, this analysis would significantly increase the contribution to the literature.

I recommend looking at Gradient Boosting in crowd ensembles for Q-learning... by Elliott et al or "The wisdom of the crowd: reliable deep reinforcement learning through ensembles of Q-functions" in TNNLS journal.  I think the work proposed in this paper is similar would work in areas where this work may not.  May be a nice item of related work.

**Summary Of The Paper:**

The authors explore a listwise action space RL approach to tackle situations with enormous, discrete action spaces or complex continuous action spaces.  Complex continuous action spaces are defined as those which have high variability in the action dimensions or have a convoluted action space consisting of a large number of features with relatively little power.

The approach utilizes stacked actors and stacked critics.  The stacked actors produce a list of actions.  The stacked critics utilize the outputs of all preceding actors when creating a training signal for the actors.

The list of actions created by the actors are candidates evaluated by a selection network.  The action selected by the selection network is the selected action for the ensemble.

**Summary Of The Review:**

An interesting approach which addresses a real-world problem with RL (although somewhat abstract).  Missing some analysis that would help justify the need for this approach.

---

> ### Author Response · Authors · 2022-11-19
> **Added More Analyses and Explained Assumptions (Part 1 of 2)**
>
> We thank you for your helpful and detailed feedback. We appreciate your comments on the importance of the real-world problem, the approach being novel and intriguing, and the reproducibility of the implementation. We agree with your suggestions, and have incorporated them as below:
>
> ### Missing some analysis to justify the need for our approach
> Value-based approaches like DQN do not scale to innumerable action spaces. We demonstrate this in Fig.5 and Appendix C.2, where we analyze the memory inefficiency of Oracle DQN w.r.t FLAIR. We vary action space dimensionality ([16, 32, 48, 64]) and action-set size ([5k, 10k, 50k, 100k, 150k]) on RecSim. DQN goes Out-of-Memory at 100k actions, while FLAIR is only at 20% GPU usage. Since the selection network in FLAIR only evaluates a small number of candidate actions (3 for RecSim), its memory requirements are much lower than DQN where Q-network enumerates and evaluates all actions. Thus, retrieval-selection frameworks are essential in practical applications where the action space grows rapidly and all the neural networks are at a large practical scale.
> Among retrieval-selection framework, prior approaches suffer from learning in complex action spaces, as they assume smoothness in the action representation space with respect to the task. Thus, a novel flexible action retrieval approach (ours) is needed. We have added this discussion in Sec 3.3.
>
> ### Explain assumption: Is the selection critic better at modelling action space complexities than the retrieval actor?
> We have added this discussion in Sec.3.3. To answer this question, the selection Qnetwork takes as input the state and an action to compute its value as Q(s, a). By taking action as an input and being trained to minimize the Bellman error, it can learn a new smoother latent space suitable for the task, where discontinuities are separated out. In prior works, Florence et al. (2022) show that models that take variables as input, better model complex discontinuous functions than those that output in the variable space. Jain et al. (2020) show that a flexible policy architecture that takes state and action representations outperforms policies that output in the action space. Thus, the selection phase is not susceptible to complex unsmooth action spaces. Hence, we focus on improving the action retrieval phase since it must output in complex action spaces.
>
>
> ### Role and Analysis of Cascaded Critics
> - **Role of Cascaded Critics in FLAIR**: We appreciate your insight on the cascaded architecture viewed from an ensembling perspective. We have improved the clarity in the description of FLAIR in Sec.4.2 and included a discussion justifying why we cascade the critics there. In brief, the cascaded critics are supposed to provide the right gradients on the list-action generated by the cascaded actors. Thus, we need paired critics to their actors, so that they can evaluate the value of an actor’s action *in the context of the currently built list of actions*. We add new experiments in Section 6.5.1 justifying the need of cascaded critics by comparing it to alternatives such as a single shared critic and a joint critic.
> - **How other actions in the list-action affect the critic output?**: In the cascaded architecture of FLAIR, the output of i-th actor is fed into the i-th critic along the state and the previous i-1 actions. Thus the i-th critic will evaluate how good of an addition an action is, to the list made of the previous i-1 actions.
> - **How does the list-action drive the ensemble to have more diversity?**: The key benefit of FLAIR is flexibility in the kinds of actions it can retrieve. The list-action is not necessarily diverse. The listwise RL agent is trained over the environment reward and learns to produce the best set of candidates for the selection Q-network. Diversity is just one of the objectives that RL reward can make this listwise procedure achieve. It can and does happen during the course of training that the listwise agent sometimes predicts a list of actions that are similar, so as to finely optimize the reward. We add qualitative results in Figure 6a, where Mine World results show FLAIR’s capability to retrieve diverse optimal actions, as this is considered useful for the task.
> - **Design Choices** and **Empirical evidence of Design Choice**: We add Sec 6.5.1 experiments, where we validate our design choice on the cascaded-critics by comparing them to two variations; (i) *Single-Critic* that trains a shared extra critic evaluating cascaded actors individually like DDPG2 k-NN, and (ii) *Joint-Critic* that trains a single joint critic evaluating cascaded actors’ output as a whole at once. As a result, we confirmed that Cascaded-Critics of ours consistently outperforms the others in both MineWorld and RecSim environments.

---

> ### Author Response · Authors · 2022-11-19
> **Added Related Work (Part 2 of 2)**
>
> ### Additional Related Work
> We appreciate the relevant works you shared that shed light into the sample efficiency of FLAIR and its relation to ensemble networks. We have added this discussion to Sec 2.3.
>
> [References]
> - [1] Florence, Pete, et al. "Implicit behavioral cloning." Conference on Robot Learning. PMLR, 2022.
> - [2] Generalization to New Actions in Reinforcement Learning, Ayush Jain, Andrew Szot, Joseph Lim Proceedings of the 37th International Conference on Machine Learning, PMLR 119:4661-4672, 2020.
> - [3] Elliott, Daniel L., K. C. Santosh, and Charles Anderson. "Gradient boosting in crowd ensembles for Q-learning using weight sharing." International Journal of Machine Learning and Cybernetics 11.10 (2020): 2275-2287.
> - [4] Elliott, Daniel L., and Charles Anderson. "The wisdom of the crowd: reliable deep reinforcement learning through ensembles of Q-functions." IEEE transactions on neural networks and learning systems (2021).

---

> ### Author Response · Authors · 2022-12-05
> **Response reminder, thank you.**
>
> Hello Reviewer CZhi, this is a gentle reminder that we have [responded](https://drive.google.com/file/d/1jKkLqk7-9o9bE6c6-l8XyWqlTGz9C3VL/view?usp=share_link) to your review. Thanks to your comments, we have made several changes:
> 1. **Justify need for our approach** $\rightarrow$ Added Fig.5 and Appendix C.2 to show that DQN does not scale to innumerable action spaces.
> 2. **Explained assumption** $\rightarrow$ Sec.3.3 (Problem Formulation) discusses why complex action space is a problem for the retrieval actor, but not the selection critic.
> 3. **Role of Cascaded Critics** $\rightarrow$ Clarified Sec 4.2 (Approach) and added new experiments in Sec 6.5.1 to explain and justify cascaded critics.
>
> Thanks again for your time and comments, as they have really helped us improve the clarity on FLAIR and additional related works! We hope we have addressed all of your concerns. Please let us know if we can answer any further questions.

---

### Author Response · Authors · 2022-11-19
**Added required details, experiments, and consistent conclusions**

We thank all the reviewers for the constructive feedback and suggestions. Our key proposition that action retrieval as listwise RL was well received. The main feedback was about missing details of our method and some clarification on performance of the Oracle method(ie., DQN) on large discrete action RL tasks. We address each comment backed with new experiments wherever required:

### More details on Problem Formulation and Approach(FLAIR)
**Missing some analysis to justify the need for our approach**(R2, R3, R4): Originally Fig.4&5 did not clearly illustrate why conventional value-based approaches like DQN do not scale to innumerable action spaces. We demonstrate this with new experiments in Fig.5 and Appendix C.2, where we analyze the memory inefficiency of Oracle DQN w.r.t FLAIR. We found that FLAIR’s action retrieval was able to maintain the low memory requirements while preserving the competitive optimality compared to DQN

**More details on Approach**(R2, R3, R4): We improved the clarity of Sec.4 to make the explanation of FLAIR self-sufficient. But due to the limited space, we added more discussions regarding the network architecture and the exact algorithm in Appendix.

**Role and Design choice of Cascaded Critics**(R1): We have improved the clarity in the description of FLAIR in Sec.4.2 and included a discussion justifying why we cascade the critics there. In brief, the cascaded critics are supposed to provide the right gradients on the list-action generated by the cascaded actors. Thus, we need paired critics to their actors, so that they can evaluate the value of an actor’s action *in the context of the currently built list of actions*. We add new experiments in Section 6.5.1 justifying the need of cascaded critics by comparing it to alternatives such as a single shared critic and a joint critic.

### Experiments
**Need of more extensive evaluation**(R2, R3)
*Extensive Quantitative Results*: Firstly, in Fig.4 we ensured the full convergence of all algorithms for a fair performance comparison to FLAIR. Also, in Sec.6.4 and C.1, we conducted further experiments where FLAIR consistently outperorms DDPG2 k-NN (the strong baseline in Fig.4) across different complexities of action spaces to demonstrate FLAIR consistently outperforms the baselines. Finally, we added more experiments to justify our design choices on the cascaded architecture of FLAIR in Figure 7 and C.3.

**Extensive Qualitative Results**: We added the qualitative analysis in Fig.6a to qualitatively analyze the properties of actions retrieved by FLAIR (which learns to output a list of actions) and DDPG2 k-NN (outputs a k-neighborhood of actions). In short, we found that cascaded actors were able to provide the candidate actions consisting of multiple optimal actions that helps the selection network’s exploration in task. Also, the website(https://sites.google.com/view/complexaction?pli=1) contains qualitative results of Mining and Reacher.

**More discussion of hyper-parameter tuning**(R2): We added the discussion of the important design choices (e.g., Architecture of critic and List-encoder) in Approach (Sec 4) and experiments (Sec.6.5). We additionally added more detailed discussions regarding the algorithm in Appendix A, and architectural choices of our contribution of listwise retrieval in Appendix C.3. Finally, we improved the clarity on the discussion of the tuning of common yet important hyper-parameters across baselines in Appendix E.3.

### Other edits
We added a discussion on related work (R1) and other notational (R2) and clarity edits like title or definitions (R1,2,3,4) to ease understanding.

### Finally…
We did our best to address the main concerns raised by reviewers and we hope that these improvements will be taken into consideration. To better highlight the changes brought to our writing, they appear in <blue> in the revised version.

[R1=CZhi, R2=oShM, R3=GAf6, R4=ieEx]

---

### Decision · Program_Chairs · 2023-01-20

**Decision:**

Reject

**Justification For Why Not Higher Score:**

After discussion with the reviewers, there was a consensus that this paper is not ready for publication.  The reviewers did feel the approach has promise, but a better description of the problem setting and more careful experiments are desired.

**Justification For Why Not Lower Score:**

N/A

**Metareview: Summary, Strengths And Weaknesses:**

Summary:

This paper presents a method of using complex innumerable action spaces in RL, by cascading DDPG learners, one per dimension.  The work is presented in contrast to standard DQN approaches which struggle with large finite action spaces, or conventional policy gradient methods that rely on smoothness in the joint action space.  The paper argues that these conditions do not hold in some less conventional RL problems (such as providing a slate of choices in a recommender system, or unfavorable formulations of Mujoco control problems).  The paper shows the proposed method does have traction on these unconventional problems.

Strengths:

The method of cascading DDPG learners appears new, and the experimental results are promising on the chosen problems.  The overall approach is simple enough to understand so that others can build upon it.  The reviewers also appreciated that the paper tackled complex action spaces in RL.

Weaknesses:

Major concerns were around the main claims of the paper and the lack of care around words that have a common technical meaning in the RL community.  Additional concerns were raised by around the lack of care in the choice and description of experiments.

The paper's main claim was of contributing complex innumerable action spaces ("Our primary contribution is introducing the problem of complex innumerable action spaces in reinforcement learning"). However, these terms were not formally defined.  Moreover using the informal problem description in the text, this is also an over-claim since prior work (not discussed in the review period) also tackles what seems to be on the surface as complex innumerable action spaces (e.g. Metz et al (2017) https://arxiv.org/abs/1705.05035).  That work went in a different direction, by discretizing continuous spaces with a sequential approach.  It would be good to have a technical definition to know what is claimed or not claimed about when the method presented in this paper is expected to work.

Another concern was on the misuse of technical terms (oracle when the method is intended to outperform the oracle, the use of exploration and robustness).  Another concern was the choice of experimental domains where all actions can be enumerated (so DQN still works adequately, so the experiment does not demonstrate the intended problem setting).

Finally, the experiment description in the original paper was found to be lacking.  Reviewers felt they had inadequate insight into when the proposed method is expected to perform well or poorly.  The experiments also raise concerns. The use of 3 seeds is poor statistical practice and computing a variance over 3 seeds is troubling.  There are no error bars on the reacher example.


**Summary Of Ac-Reviewer Meeting:**

All the reviewers met and discussed the paper.

The main point in favor for the paper was that the reviewers appreciated tackling more complex action spaces in RL. The reviewers noted the method could demonstrate performance on some domains, and the method might transfer to some technologically relevant domains.

However, the consensus was that even after the author response and paper revision, the paper fell short in multiple ways.   The claimed contribution was deemed insufficiently clear, the technical exposition was not sufficiently careful, the motivation for the cascades was still lacking, and the experiments were not adequately well chosen, analyzed, or discussed.